# Abnormal proliferation of gut mycobiota contributes to the aggravation of Type 2 diabetes

Li Bao[1,2], Ying Zhang[2,3], Guoying Zhang[4], Dechun Jiang[1,2] & Dan Yan [2,3✉]

Type 2 diabetes (T2D) constitutes a worldwide health threat, and the underlying mechanism for the development and progression of T2D is complex and multifactorial. During the last decade, gut commensal bacteria have been found to play a crucial role in the regulation of T2D and related metabolic disorders. However, as a considerable component in gut microbiome, the relationship between mycobiota and T2D and related metabolic disorders remains unclear. As a proof-of-concept, we observed that the ablation of the commensal fungi in mice can protect HFD (High fat diet) induced insulin resistance and related metabolic disorders. Both ITS2 (internal transcribed spacer 2) sequencing and culture-dependent analysis show the enrichment of *Candida albicans* in samples from individuals with T2D (Chinese Clinical Trial Registry, ChiCTR2100042049). Repopulation with *C. albicans* in HFD mice accelerated insulin resistance and related disorders. Mechanically, we found the β-glucan from *C. albicans* mirrored the deteriorating effect of *C. albicans* through the dectin-1 dependent pathway. Our current findings support that gut mycobiota play an important role in the progress of T2D and indicated the preventing of gut mycobiota is a promising strategy to alleviate insulin resistance and related metabolic dysfunctions.

[1] Department of Pharmacy, Beijing Shijitan Hospital, Capital Medical University, No.10 Tieyi-Road, Haidian District, 100038 Beijing, China. [2] Beijing Key Laboratory of Bio-characteristic Profiling for Evaluation of Rational Drug Use, No.10 Tieyi-Road, Haidian District, 100038 Beijing, China. [3] Beijing Friendship Hospital, Capital Medical University, No. 95 Yong'an Road, Xicheng District, 100050 Beijing, China. [4] Biomedical Innovation Center, Beijing Shijitan Hospital, Capital Medical University, No.10 Tieyi-Road, Haidian District, 100038 Beijing, China. ✉email: danyan@ccmu.edu.cn

Type 2 diabetes (T2D) and related metabolic disorders constitute a worldwide health threat[1], and the underlying mechanism for the development and progression of T2D is complex and multifactorial. During the last decade, gut microbiota-mainly represent by gut commensal bacteria, have been found to play a crucial role in the regulation of energy and substance metabolism, which affect the development of T2D and related metabolic disorders[2–5]. Imidazole propionates (ImP), produced by several bacterial strains, including *Streptococcus mutans* and *Eggerthella lenta*, which was found to be elevated in individuals with T2DM and impairs insulin signaling through activation of the p38γ-p62-mTORC1 pathway[6]. GUDCA, an endogenous bile acid conversed by gut bacteria, was found to be increased by metformin treatment and improve insulin sensitivity though an intestinal FXR dependent pathway[7]. However, the function and action mechanisms of remain gut symbiont on the host remain unknown.

In recent few years, as a considerable component in gut microbiome, the gut mycobiota (gut fungi) has found to have enormous impact in mammalian health and disease[8,9]. Increasing evidence suggests that commensal fungi can affect the course and severity of a several of diseases, such as asthma, inflammatory bowel disease, pancreatic carcinoma, and alcoholic liver disease[10–13]. Candidalysin, a peptide toxin secreted by the commensal gut fungus *Candida albicans*, was found to enhance alcohol-associated liver disease independent of the β-glucan receptor CLEC7A, and is also associated with worse clinical outcomes in patients with alcoholic hepatitis[14]. However, the relationship between mycobiota and other diseases, especially in T2D and related metabolic disorders remains unclear.

Herein, we observed that the ablation of the commensal fungi in mice can protect HFD-induced insulin resistance and related metabolic disorders, which indicated the potential role of mycobiota in T2D. Both ITS2 sequencing and culture-dependent analysis show the enrichment of *C. albicans* in samples from individuals with T2D. Repopulation with *C. albicans* in HFD mice accelerated insulin resistance and related disorders. Mechanically, we found the β-glucan from *C. albicans* mirrored the deteriorating effect of *C. albicans* through the dectin-1 dependent pathway. Furthermore, *C. albicans* also interact with gut bacteria, with the enrichment of *Staphylococcus xylosus* and *Mucispirillum schaedieri*. Our current findings support that gut mycobiota play an important role in the progress of T2D and indicated the preventing of gut mycobiota is a promising strategy to alleviate T2D and related metabolic dysfunctions.

## Results

### Eliminate of gut mycobiota alleviated insulin resistance and related metabolic disorders in HFD-fed mice.

To evaluated weather gut mycobiota play a role in the occurrence and progress of T2D and related metabolic disorders, three most widely used antifungal agents for gut mycobiota research, including amphotericin B, fluconazole, and 5-fluorocytosine[15,16], were dissolved in drinking water with a final concentration of 100 mg L$^{-1}$, 500 mg L$^{-1}$, and 1 g L$^{-1}$ and given to HFD mice for determine if eliminate of the intestinal fungal community impacts the progress of insulin resistance and related disorders. As the diet control, the standard diet fed (CD) mice was performed.

Firstly, all the antifungal agents eliminate the fungi effectively, as indicated by qPCR (Supplementary Fig. 1). Sequential monitoring of the blood glucose showed that antifungal agents improved fasted blood glucose and free-diet blood glucose significantly (Fig. 1a, b), but they did not showed effect on the cumulative food intake (Fig. 1c). Insulin resistance is a central pathological factor in T2D and a causal link between T2D and

related metabolic disorders, including obesity, hyperlipidemia, and nonalcoholic fatty liver disease (NAFLD)[17]. HFD mice displayed apparent insulin resistance (Fig.1d–g). Antifungal agents treatment significantly decreased the HOMA-IR (Fig. 1d). Moreover, in the oral glucose tolerance test (OGTT), HFD mice treated by AmpB, FCZ, and 5FU showed decrease of blood glucose levels at 15 and 30 min compared with the HFD mice (Fig. 1e). The AOC of the Antifungal agent groups were much lower than that in the HFD group (Fig. 1f). Above insulin sensitizing effects also proved by insulin tolerance test (ITT) test (Fig. 1g), which together indicated the role of gut intestinal fungal community in the control of blood glucose level and insulin resistance.

Obesity, dyslipidemia, and NALFD is the most common T2D related disorders with severely impair on human health. Compared to the vehicle-treated HFD-mice, supplementation with antifungal agents significantly prevented body weight gain at 2-week treatment, and extend to the end of treatment (Fig. 1h). Regarding to the blood dyslipidemia, antifungal agents treated HFD-mice effectively improved blood dyslipidemia by reducing the concentrations of plasma total cholesterol (TC), low-density lipoprotein cholesterol (LDL-C), and triglycerides (TGs) (Fig. 1i, j). Moreover, the size of adipocytes in the white adipose tissue (WAT) of antifungal agents treated HFD-mice was reduced (Fig. 1o).

In terms of NAFLD, we noticed the recovery of liver weight to body weight ratio treated by antifungal agents (Fig. 1k) and the improvement of markers of hepatocellular injury, such as plasma alanine aminotransferase (ALT) and aspartate aminotransferase (AST; Fig. 1l). In addition, antifungal agents treated HFD-mice apparently reduced hepatic macrosteatosis, hepatocyte ballooning, and fat deposition, as indicated by liver sections HE and Oil O red staining and microscopy observation (Fig. 1o). Furthermore, antifungal agents supplement reverse systems inflammation, a causal link between insulin resistance, obesity and diabetes, as indicated by the alleviated plasma TNF-α and LPS level (Fig. 1m, n).

Overall, our results found that the gut mycobiota eliminate treated by three antifungal agents have profound impact on T2D, insulin resistance and related disorders, indicated that gut mycobiota may play an important role in the progress on T2D and related disorders.

### Gut mycobiota alternations in T2D patients.

For further investigate the relationship between gut mycobiota and T2D, we collected stool samples from 11 individuals with type 2 diabetes and 6 healthy volunteers. The over-growth of mycobiota in T2D group was observed by real-time RT-PCR (Fig. 2a). Further, we performed ITS2 sequencing to explore the composition and abundance of gut mycobiota between T2D and control group. There is no significant difference between two groups for α diversity, which indicated by OTU number and Shannon index (Fig. 2b, c). The β diversity show the composition and abundance of microbiota from T2D group was significantly differed from that of control group (NMDS plot, MRPP test, $p < 0.001$; Fig. 2d).

Among the fungal taxa, the phylum Ascomycota dominated the mycobiota, while Basidiomycota was observed as the second most abundant phylum (Fig. 2e) in both T2D and control groups. Additional differences were observed at lower taxonomic levels, at the genus level, 6 fungal features were enriched in T2D, including *Candida*, *Aspergillus*, *Sarocladium*, *Debaryomyces*, *Kurtzmaniella* and *Monascus*, whereas the genus including *Fusarium* and *Cladosporium* were significantly reduced in T2D patient (Fig. 2f). The analysis with linear discriminant analysis (LDA) effect size (LEfSe) method revealed a significant increase in *Candida albicans*, Saccharomycetales in T2D patient, accompanying the

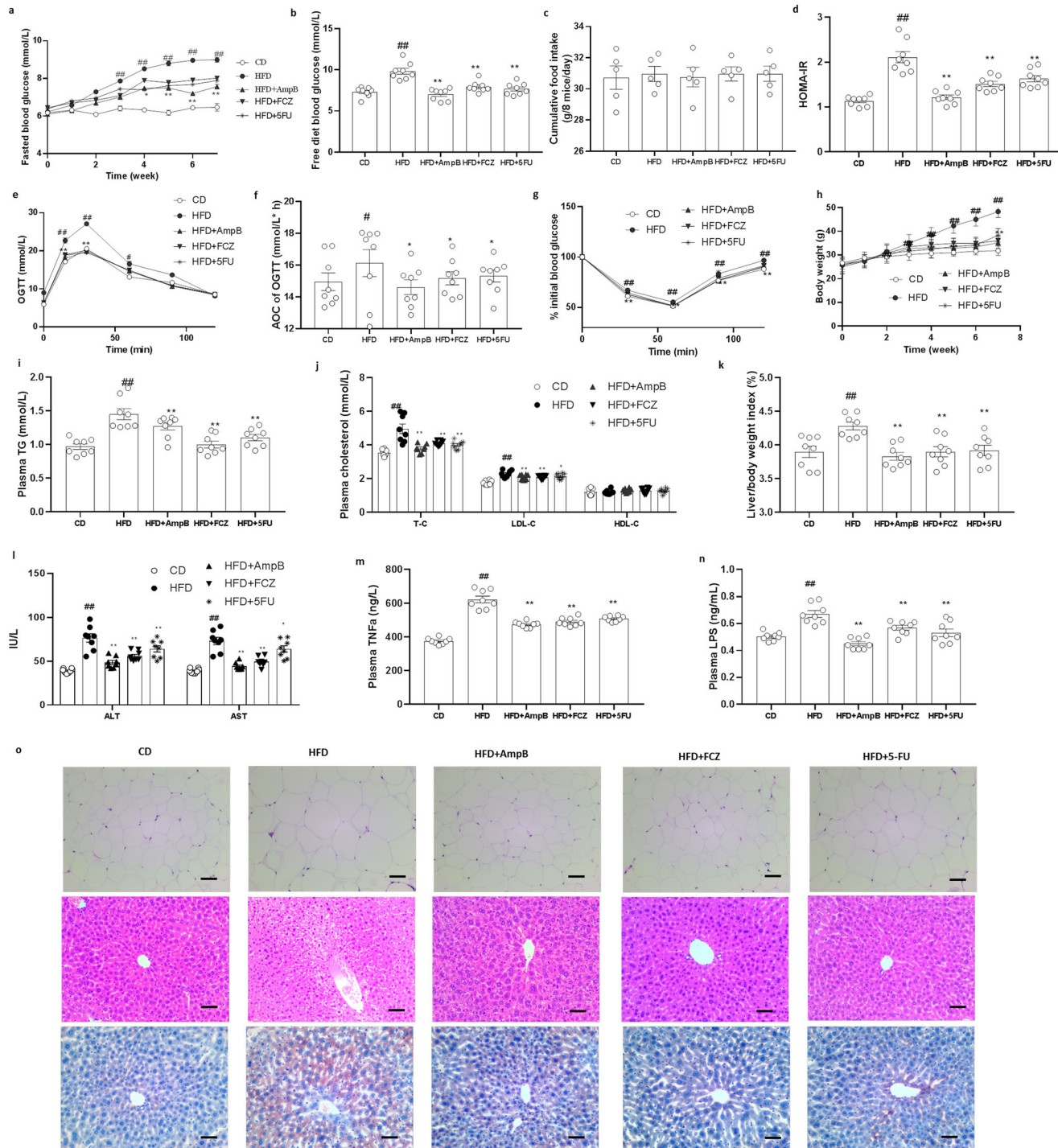

**Fig. 1 Antifungal agents treatment alleviated glucose metabolism related disorders in HFD-fed mice. a** Fasted blood glucose, **b** Free blood glucose (6th week), **c** Cumulative food intake (5days recorded), **d** HOMA-IR, **e** OGTT and **f** AOC, **g** ITT, **h** Weight change, **i** Plasma triglyceride, **j** Plasma total cholesterol, LDL-C and HDL-C, **k** liver/body weight ratio, **l** plasma alanine trasaminase (ALT) and aspartate transminase (AST), **m** Plasma TNF-α, **n** Plasma LPS, **o** representative H & E-stain pictures of WAT tissue, and representative H & E-stain and oil-red stain pictures of liver tissue (Scale bars, 100 μm ($n = 3$ mice per group)). Data are presented as the mean ± standard error of the mean (SEM); $n = 8$ mice per group. Statistical analysis was done using one-way ANOVA followed by the Tukey post hoc test. $^*P < 0.05$; $^{**}P < 0.01$ vs HFD group, and $^\#P < 0.05$; $^{\#\#}P < 0.01$ vs CD group.

decrease of Hypocreales, Sordariomycetes, and Nectriaceae (Fig. 2g, h).

While sequencing-based profiling is useful for providing a snapshot of fungal DNA, whether the associated organisms are passive bystanders dead upon arrival or viable inhabitants of a given niche cannot be reliably determined. We aimed, therefore, to determine if the identified sequences came from viable

organisms via cultivation, and if so, we hypothesized that the genomic and functional characteristics of these organisms could provide insights into the micro-environment of gut mycobiota. In according with sequencing results, we discovered a significant enrichment of *Candida albicans* in the T2D group (Fig. 2i), which indicated *C. albicans* as a potential pathogens in the pathological process of insulin resistance and diabetes.

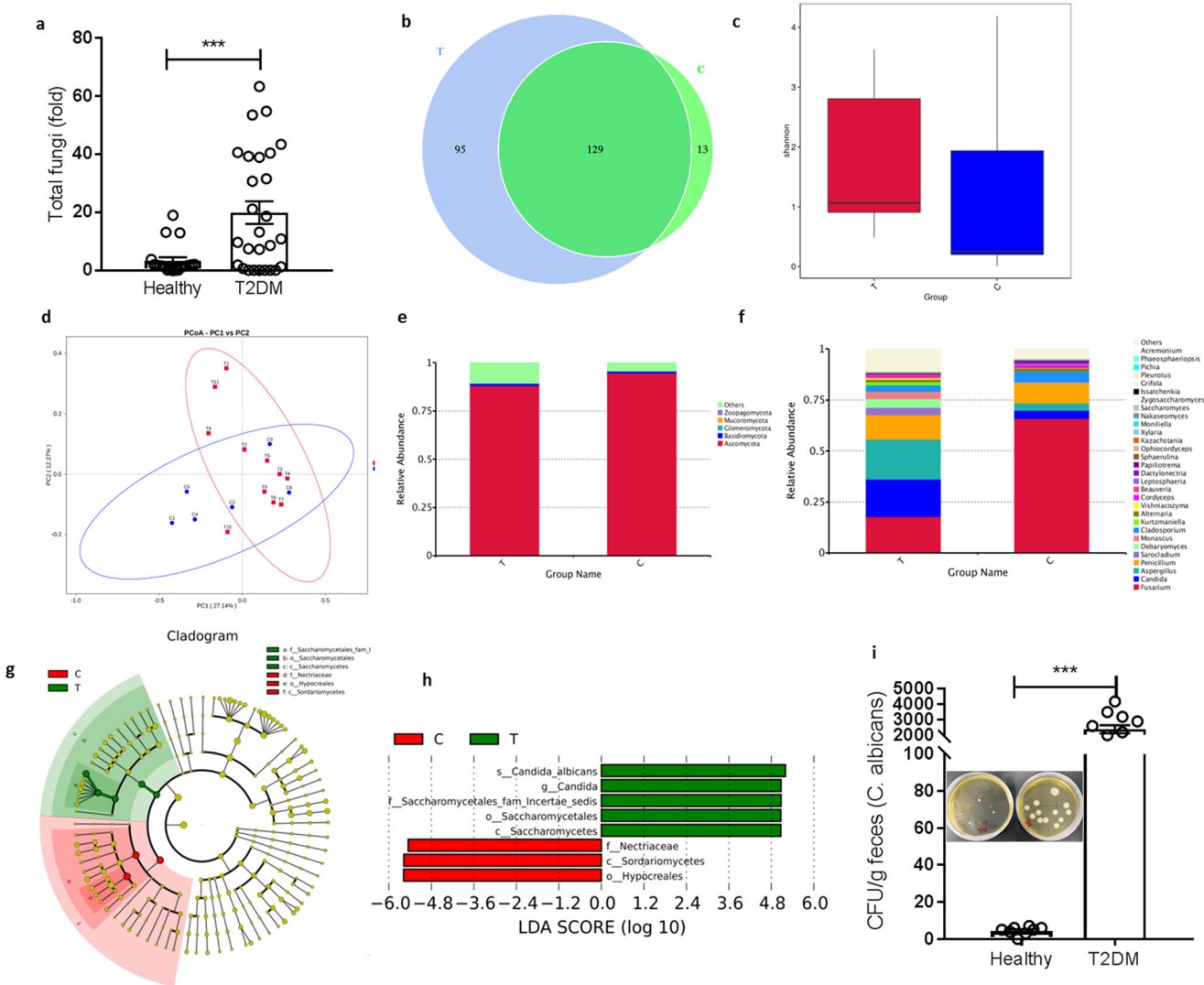

**Fig. 2 Gut mycobiome (fungal community) alternations in T2D patients. a** Total fungi in feces were assessed by qPCR (healthy group: $n = 20$, T2DM group: $n = 27$; ***$p < 0.001$ vs healthy group), **b** Venn diagram based on genera, **c** Shannon index, **d** Unweighted uniFrac-based principal coordinates analysis, Change in fungi at the phylum (**e**) and genus **f** levels, **g** Cladogram generated from LEfSe analysis showing the relationship between taxon, **h** Linear discriminant analysis (LDA) scores derived from LEfSe analysis, **i** CFUs of *C. albicans* in fecal samples (healthy group: $n = 8$ persons, T2DM group: $n = 12$ persons; ***$p < 0.001$ vs healthy group). T: T2DM patients, C: healthy volunteers.

***C. albicans* accelerated the progress of T2D and related disorders.** *Candida albicans*, a prominent opportunistic pathogen responsible for mucosal infection in human immunodeficiency virus-infected patients and nosocomial systemic infections, is now regarded as common gut symbionts colonize all segments of the human digestive tract[18]. *C. albicans* is found to increase in patients with alcoholic hepatitis and is the most abundant *Candida* species in patients with end-stage liver disease due to alcohol abuse[19]. However, the precise role of *C. albicans* in T2D and related disorders remains unknown. Thus, we repopulated CD and HFD-mice with *C. albicans*, in order to explore the effects of *C. albicans* in insulin resistance and related metabolic disorders.

In comparison with the vehicle group, *C. albicans* significantly accelerated the features of T2D and insulin resistance in the HFD-fed mice, indicated by the increasing fasted blood glucose, HbA1c level and HOMA-IR (Fig. 3a–d), and the impairing insulin tolerance and glucose tolerance (Fig. 3e–h). There has no significant change on body weight, liver/body weight index, and plasma lipid profiles during *C. albicans* colonization (Supplementary Fig. 2). Moreover, the H&E and Oil O red staining also

proved the aggravating macrosteatosis and hepatocyte ballooning in *C. albicans* group (Fig. 3i). However, the effects of *C. albicans* on metabolic disorders were disappeared in CD-fed mice (Fig. 3).

**The structural characterization of β- glucan from *C. albicans*.** We next explored the molecular basis of the insulin resistance inducing effect in *C. albicans*. 1,3-β-glucans-a cell wall polysaccharides found in and released from most fungi, including species of *Candida*, *Aspergillus*, and *Pneumocystis*[20], were found to make a profound impact on host, including immune response, hepatic injury, carcinogenesis[12,13,21]. Accordingly, patients and mice with T2D has a higher level of 1,3-β-glucans in both fecal and plasma (Fig. 4a, b), indicated the potential role of 1,3-β-glucans on the occurrence of T2D.

Due to the species variety, chemical structure and bio-function of 1, 3-β-glucans may be widely divergent. For example, the 1, 3-β-glucans derived from *Aspergillus* spp. may induce strong immune response and trigger asthma[10], on the contrary, the 1,3-β-glucans isolated from medical fungi *Poria cocos* exhibit a

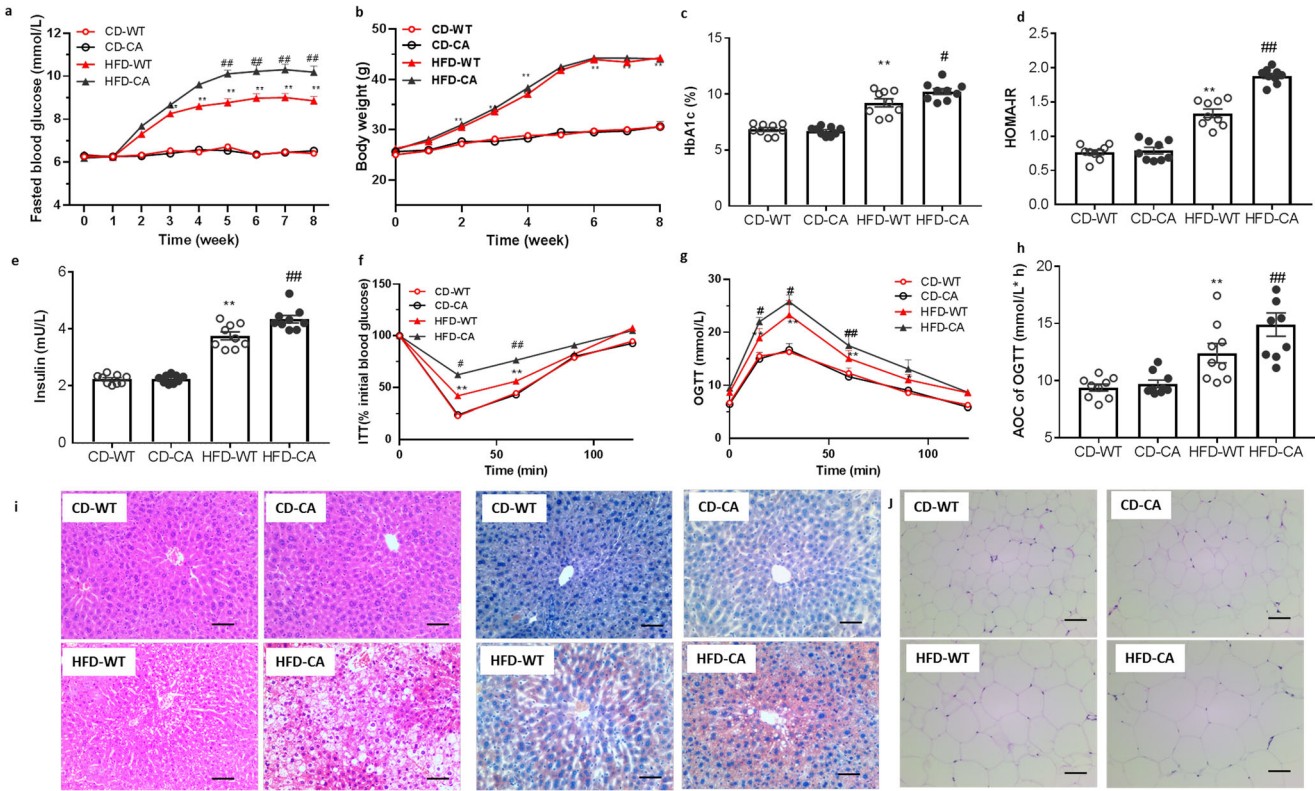

**Fig. 3 Effects of *C. albicans* strains isolated from T2DM patients on the induction of insulin resistance.** In both CD fed and HFD fed mice. **a** fasted blood glucose, **b** Body weight, **c** HbA1c, **d** HOMA-IR, **e** insulin level, **f** ITT on the 29th day, **g** OGTT, and **h** AOC on the 36th day of treatment, **i** representative images of H & E-stain, oil-red stain of liver tissue, and **j** representative images of H & E-stain of adipose tissue, Scale bars = 100 μm (*n* = 3 mice per group). Data are presented as the mean ± standard error of the mean (SEM); *n* = 9 mice per group. Statistical analysis was done using one-way ANOVA followed by the Tukey post hoc test. *$P < 0.05$; **$P < 0.01$ vs CD-WT group, and #$P < 0.05$; ##$P < 0.01$ vs HFD-WT group.

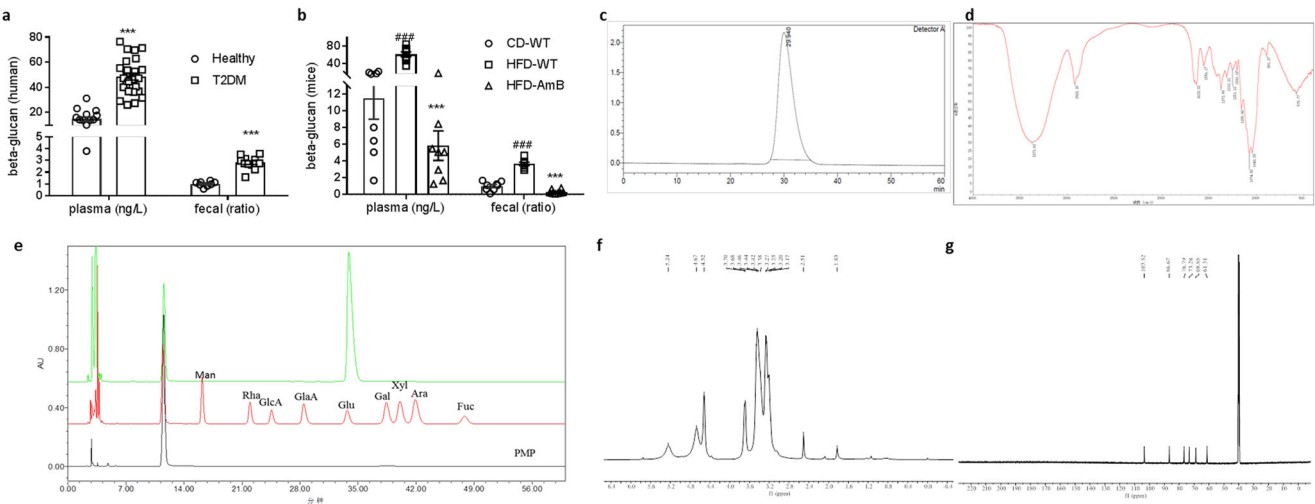

**Fig. 4 The characterization of 1,3-β-glucan isolated from *C. albicans*.** 1,3-*β*-glucans content of plasma (healthy group: *n* = 15, T2DM group: *n* = 26) and fecal (healthy group: *n* = 8, T2DM group: *n* = 12) in human (**a**) and mice (**b**, *n* = 8), **c–g** HPGPC, FT-IR and NMR in DMSO-*d₆* spectrum of 1,3-β-glucan isolated from *C. albicans*. **e** The monosaccharide composition of CAG was determined by HPLC, red line: a standard mixture, green line: CAG acid-hydrolyze and black line: PMP derivate. Statistical analysis was done using one-way ANOVA followed by the Tukey post hoc test. ***$P < 0.001$ vs healthy group (A) and ***$P < 0.001$ vs HFD-WT group, and ###$P < 0.001$ vs CD-WT group.

protective effect on metabolic syndrome[22,23]. In order to explore the effect of β-glucan from *C. albicans* (CAG) on insulin resistance and related metabolic disorders, we next identify CAG using a *C. albicans* strain isolated from T2D patient. High-performance size exclusion chromatography analysis of CAG indicated a single homogeneous composition (Fig. 4c). The

Fourier transform infrared spectrum (FT-IR) spectrum of CAG showed the absorptions at 3373, 2922, 1200–1000 cm⁻¹, corresponding to the vibration of O-H, C-H, C-O, C-O-C bonds, respectively. The absorptions at 1074 and 891 cm⁻¹ suggested a pyranose form of sugars with β anomeric configuration (Fig. 4d). The monosaccharide composition of the polysaccharide was

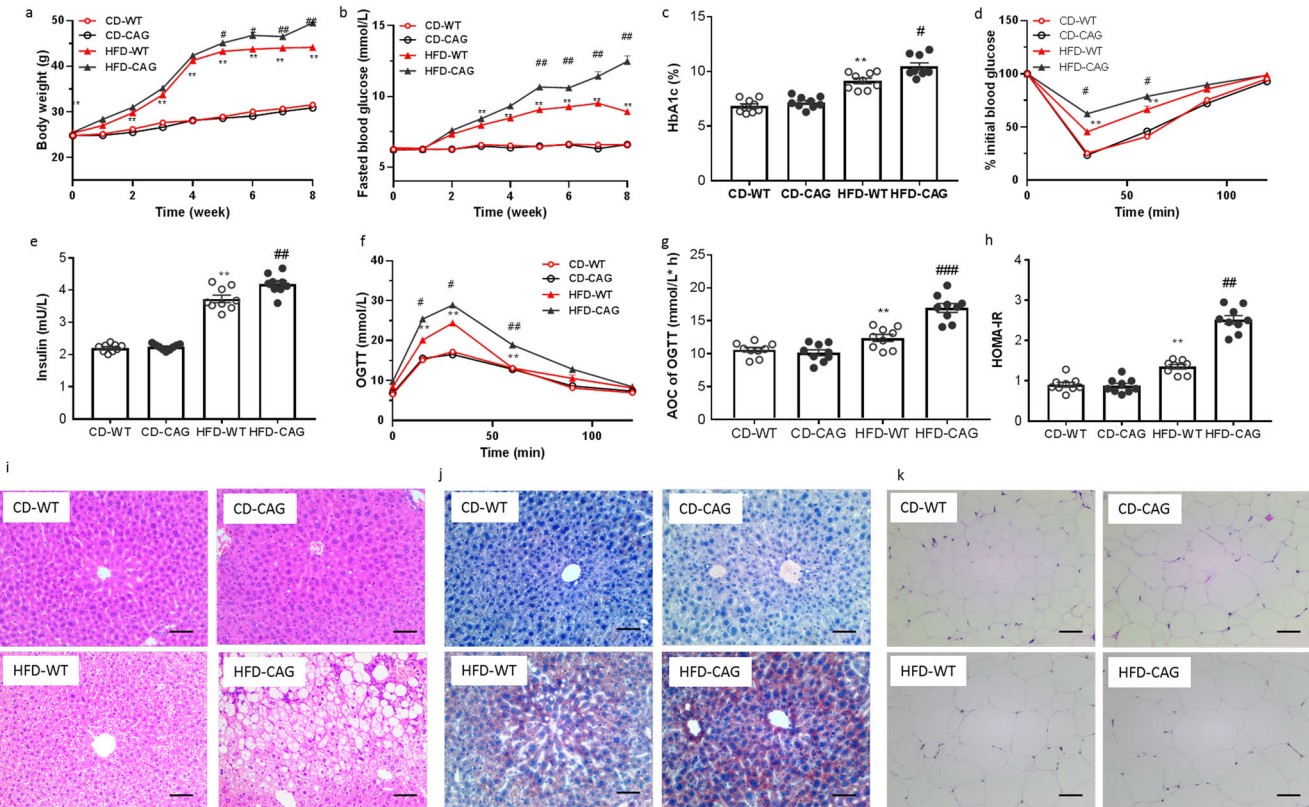

**Fig. 5 Feeding with 1,3-β-Glucan isolated from *C. albicans* (CAG) promoted insulin resistance.** In both CD fed and HFD fed mice. **a** Body weight, **b** sequential monitoring of blood glucose after 4 h fasting, **c** HbA1c, **d** ITT on the 29th day, **e** plasma insulin, **f** OGTT and **g** AOC on the 36th day of treatment in HFD-fed mice, **h** HOMA-IR, **i, j** representative images of H & E-stain, oil-red stain of liver tissue, and **k** representative images of H & E-stain of adipose tissue, Scale bars = 100μm (*n* = 3 mice per group). Data are presented as the mean ± standard error of the mean (SEM); *n* = 9 mice per group. Statistical analysis was done using one-way ANOVA followed by the Tukey post hoc test. *$P < 0.05$; **$P < 0.01$ vs CD-WT group, and #$P < 0.05$; ##$P < 0.01$ vs HFD-WT group.

determined as D-glucose by acid hydrolysis followed with HPLC analysis of the corresponding derivatives (Fig. 4e). By comparing [13]C NMR spectra of CAG with the corresponding compounds in the literatures[24], confirmed the main chain of CAG to be 1,3-β-D-glucan on the basis of the signals observed at 73.3 (C-2), 86.7(C-3), 76.8 (C-5) (Fig. 4f, g).

**CAG mirrored the pathogenic effect of *C. albicans*.** As expected, HFD-feeding significantly increased both body weight and free diet blood glucose, but there is no effect on CD-WT or CD-CAG group (Fig. 5a, b). HbA1c was raised substantially (Fig. 5c). CAG also deteriorate glucose tolerance and insulin resistance in HFD mice. The AOC during the OGTT and the ITT of CAG-treated HFD mice was much higher than that of the vehicle-treated HFD mice (Fig. 5d–g). CAG treatment also caused elevation in the HOMA-IR (Fig. 5h). The H&E and Oil O red staining of liver in CAG group also mirrored the aggravating effect on macrosteatosis and hepatocyte ballooning in *C. albicans* group (Fig. 5i–k). Above observations indicated that oral CAG exacerbate insulin resistance and related disorders, mirrored the pathogenic effect of *C. albicans*. According to the effects of *C. albicans*, CAG treated CD-fed mice shows compared metabolic phenotype with CD mice (Fig. 5).

**CAG induce insulin resistance though the activation of dectin-1 pathway.** Dectin-1(also known as C-type lectin domain family 7 member A, CLEC7A) is a pattern recognition receptor, which recognizes a variety of 1, 3 -β-glucans[25]. Upon ligand binding, Dectin-1-Syk pathway activates macrophage polarization by

activating IRF5, a transcription factor that regulates proinflammatory CD11c[+] macrophage differentiation[26], which induce insulin resistance and related disorders. CAG treatment significantly induced the expression of genes including Dectin-1, Syk, and Irf5 (Fig. 6b, d and Supplementary Fig. 3b), indicate the activation of Dectin-1-Syk pathway. Accordingly, the activation of Dectin-1 signing also observed in the *C. albicans* treated HFD mice (Fig. 6a, c and Supplementary Fig. 3a).

The induced dectin-1 pathway is in line with the damaged intestinal barrier function and intestinal inflammation in both CAG and *C. albicans* treated group, compare with vehicle mice respectively, indicated by the reduced *Zo1* and *Occludin* expression (Fig. 6e, f), and the induced inflammatory factors (Fig. 6g, h). Above observation suggested that *C. albicans* and CAG both induced the Dectin-1-Syk pathway through the secretion of 1,3-β-glucans, thus aggravate the insulin resistance and related disorders.

Furthermore, we performed the dectin-1 antagonist laminarin to further verify the role of dectin-1 pathway in *C. albicans* and CAG induced phenotype. As a dectin-1 antagonist, laminarin treatment reversed the deteriorated effect of both *C. albicans* and CAG on HFD-treated mice as indicated by body weight gain, fasted blood glucose, ITT, OGTT, HOMA-IR, HbA1c (Fig. 7a–n). Thus, we concluded that *C. albicans* and CAG induce insulin resistance and related inflammation by a dectin-1 dependent pathway.

## Discussions

Current studies indicated that mycobiota are inseparable from human health. Dysbiosis and invasion of mycobiota are

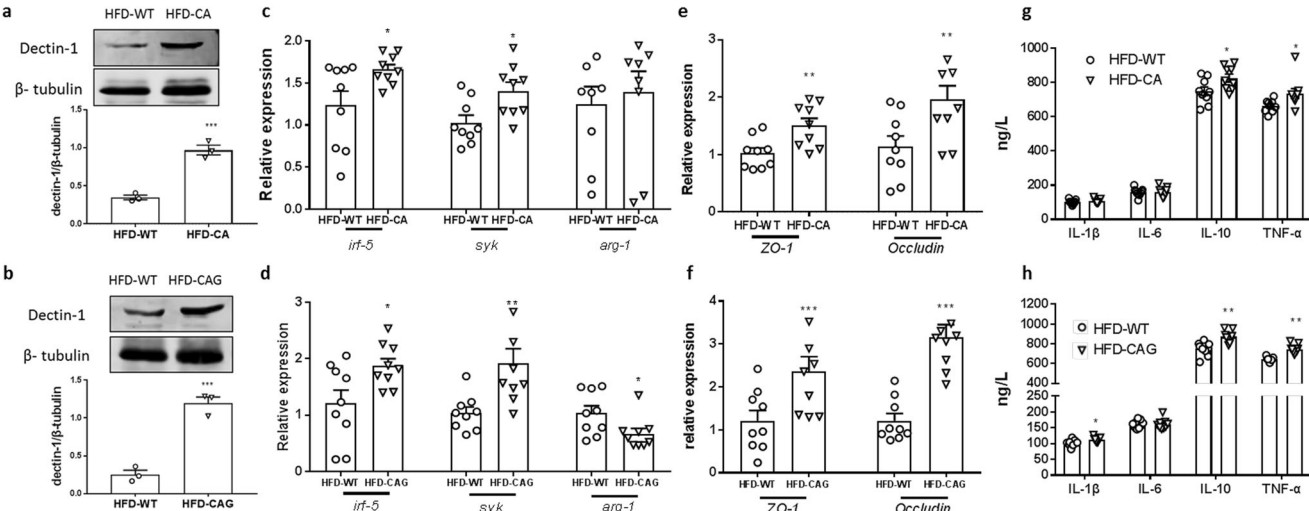

**Fig. 6 CAG induce insulin resistance though the activation of dectin-1 pathway. a**, **b** The colon dectin-1 level of the *C. albicans*-treated or CAG-treated mice (*n* = 3 mice per group), **c**, **d** The expression of *irf-5*, *syk* and *arg-1* in colon, **e**, **f** The expression of *ZO-1* and *occludin* in colon, **g**, **h** The serum levels of IL-1β, IL-10, IL-6, and TNF-α. Data are presented as the mean ± standard error of the mean (SEM); *n* = 9 mice per group. Statistical analysis was done using one-way ANOVA followed by the Tukey post hoc test. *P < 0.05; **P < 0.01 and ***P < 0.001 vs HFD-WT group.

**Fig. 7 Dectin-1 Antagonist Laminarin treatment reversed the deteriorated effect of *C. albicans* and CAG on HFD-treated mice.** Effects of Laminarin on C. albicans induced metabolic disorders. **a** Body Weight, **b** Fasted blood glucose, **c** ITT on the 31th day, **d** OGTT and **e** AOC on the 34th day, **f** HOMA-IR, **g** HbA1c. Effects of Laminarin on CAG induced metabolic disorders. **h** Body Weight, **i** Fasted blood glucose, **j** ITT on the 31th day, **k** OGTT and **l** AOC on the 34th day, **m** HOMA-IR, **n** HbA1c. Data are presented as the mean ± standard error of the mean (SEM); *n* = 9 mice per group. Statistical analysis was done using one-way ANOVA followed by the Tukey post hoc test. *P < 0.05; **P < 0.01 and ***P < 0.001vs HFD-CA (CAG) group, and #P < 0.05; ##P < 0.01 vs HFD-WT group.

confirmed to cause disease in different parts of the body[27]. The role of fungi in intestinal inflammation and many gastrointestinal and liver diseases have come up. *C. albicans* can worsen colitis in mice and has an increased abundance in patients with Crohn's disease, although the pathogenic mechanisms are not yet fully understood[28]. Aykut et al. show that fungi migrate from the gut lumen to the pancreas, and that this is implicated in the pathogenesis of pancreatic ductal adenocarcinoma (PDA)[13]. In a study of patients with alcoholic liver disease (ALD), gut fungal load, primarily *Candida* spp., was increased compared with healthy control individuals[20]. However, the role of mycobiota in diabetes and insulin resistance has not been clearly implicated. In current prove-of-concept study, by eliminating the commensal fungi in mice, we observed the potential role the mycobiota in diabetes and insulin resistance, which raising the pathogenic potential of misadjusted mycobiota in diabetes and related metabolic disorders.

*Candida albicans* is a prominent opportunistic pathogen and now regarded as a common gut symbionts colonize all segments of the human digestive tract[18]. Patients with alcohol use disorder have higher intestinal proportions of *C. albicans*. Candidalysin, a cytotoxic peptide secreted by *C. albicans*, causes direct hepatocyte damage and exacerbates ethanol-induced liver disease in mice[14]. Otherwise, commensal *C. albicans* also play a protective role in several special situation. Shao et al. show commensal *C. albicans* uniquely activates circulating immune cells to protect against systemic infection by invasive extracellular microbial pathogens[29]. Thus, the role of commensal *C. albicans* in diabetes need be established. In current study, combine with high-throughput sequencing and fungal cultivation of clinical samples, single fungi gavage and pathway analysis, we conclude that the β-glucan from common symbiotic fungus *C. albicans* accelerated the progress of T2D and related disorders though the activation of dectin-1 pathway. Above evidence revealed the pathogenic role of *C. albicans* in diabetes and also highlight this microbe as a novel target for the clinical treatment of T2D; that is, developing a 'drugs for bugs' approach in the management of this common disease.

Many observations suggest that bacteria and fungi interact within the gut, influencing each other through different levels of symbiosis. However, most of the data on fungi-bacteria interactions have been collected in other settings, such as the lung and oral or vaginal cavities[30]. By the 16S rRNA amplicon profiles, we showed that the *C. albicans* gavage enhance the relative abundance of genus *Staphylococcus*, *Mucispirillum* and decrease genus *Akkermansia*. *Staphylococcus* spp. may produce mounts of LPS, a cell wall component of Gram-negative intestinal bacteria, can enter the blood stream through the impaired barrier of the gut mucosa, thus induce chronic low-grade inflammation (CLGI), a key mediator of the development of obesity, insulin resistance, and diabetes[31–33]. By the formation of polymicrobial biofilm, increased *C. albicans* can promote the growth of *Staphylococcus* spp., thus enrich plasma LPS concentration and aggravate insulin resistance. *Akkermansia Muciniphila*, a bacterium widely present in the intestinal tract of humans and other animals, is regarded as a core functional microbe with prominent beneficial effects in lipid metabolism disorders and many other diseases[34–36]. The increased *A. Muciniphila* induced by *C. albicans* indicated the potential antagonism, which may explain the extensive role of commensal *C. albicans* in host physiological status. Further studies need carry out to explore the underlying mechanism between *A. Muciniphila* and *C. albicans*.

In conclusion, this is a prove-of-concept study to explore the role of mycobiota in diabetes and related metabolic disorders. By the ablation of the commensal fungi in mice, we observed the protection of HFD-induced insulin resistance. Both ITS2 sequencing and

culture-dependent analysis show the enrichment of *C. albicans* in samples from individuals with newly diagnosed T2D. Repopulation with *C. albicans* in HFD mice accelerated T2D and related disorders. Furthermore, we found the β-glucan from *C. albicans* mirrored the deteriorating effect of *C. albicans* through the dectin-1 dependent pathway. *C. albicans* also interact with gut bacteria, with the enrichment of *Staphylococcus xylosus* and *Mucispirillum schaedieri*. Our current findings support that gut mycobiota play an important role in the progress of T2D and indicated that the improvement of *C. albicans* is a promising strategy to alleviate T2D and related metabolic dysfunctions.

## Methods

**Animal care and experiments**. All animal experiments were approved by the Ethics Committee of Beijing Shijitan Hospital, Capital Medical University (permission No. 2017-keyanlunshen-35). 6-week-old C57BL/6J male mice were purchased from Beijing Vital River Laboratory Animal Technology Co., Ltd. (Beijing, China). For chow-diet fed, mice were fed with a rodent diet with 10 kcal% fat (Cat. D12450J, Research Diet, New Brunswick, NJ, USA), and for HFD mice, mice were fed with a rodent diet with 60 kcal% fat (Cat. D12492, Research Diet, New Brunswick, NJ, USA)

For antifungal treatment assay, 6-week-old C57BL/6J male mice were divided into 5 groups ($n = 8$) randomly: standard diet fed (CD-WT) and high-fat diet (HFD-WT) were given distilled water, and the 3 other groups were treated with sterile water supplemented with antifungal drugs amphotericin B (HFD + AmB, final concentration: 100 mg/L), fluconazole (HFD + FCZ, final concentration: 500 mg/L) and 5-fluorocytosine (HFD + 5-FU, final concentration: 1 g/L) for 7 weeks. For *C. albicans* treatment assay, *C. albicans* isolated from T2DM patients were cultivated on Yeast peptone dextrose Agar (YPD) and potato dextrose agar (PDA, Supplementary Table 1) for 5–7 days and then propagated in potato dextrose broth (both at 28 °C) prior to oral gavage into mice. To prove the effect of *C. albicans* (CA), C57BL/6 male mice were sorted into 4 groups: CD-WT (standard diet fed + potato dextrose broth), CD-CA (standard diet fed + oral administration of *C. albicans* at the dose of $2 \times 10^7$/day for the following 7 weeks), HFD-WT (HFD diet + potato dextrose broth), HFD-CA (HFD diet + oral administration of *C. albicans* at the dose of $2 \times 10^7$/day for the following 7 weeks).

To prove the effect of 1,3-β-glucan on HFD-induced T2DM, C57BL/6 male mice were sorted into 4 groups: CD-WT (standard diet fed + sterilized water), CD-Glucan (standard diet fed + oral administration of 1,3-β-glucan at the dose of 100 mg/kg/day for the following 7 weeks), HFD-WT (HFD diet + sterilized water), HFD-Glucan (HFD diet + oral administration of 1,3-β-glucan at the dose of 100 mg/kg/day for the following 7 weeks).

To test whether the aggravation of *C. albicans* and CAG on T2DM is depended on dectin-1, dectin-1 antagonist laminarin (Lam) was used. C57BL/6 male mice were sorted into 3 groups: HFD-WT (HFD diet + sterilized water), HFD-CA-CA-Lam (HFD diet + *C. albicans* at the dose of $2 \times 10^7$/day + Laminarin at the dose of 250 mg/kg/day for the following 7 weeks). Meanwhile, Another C57BL/6 male mice were sorted into 3 groups: HFD-WT (HFD diet + sterilized water), HFD-CAG-CA-Lam (HFD diet + CAG at the dose of 100 mg/kg/day + Laminarin at the dose of 250 mg/kg/day for the following 7 weeks).

**Tissue sampling**. After treatment, animals were anesthetized with diethyl ether and blood was sampled from the portal and cava veins. After exsanguination, mice were euthanized by cervical dislocation. Abdominal white adipose tissue (WAT), intestines, and liver were precisely dissected, weighed, immediately immersed in liquid nitrogen, and stored at −80 °C for further analysis.

**Biochemical analyses**. Metabolic indexes were performed as reported[37], details as follows: serum glucose, serum insulin, triglyceride (TG), and total cholesterol (TC), high density lipoprotein cholesterol (HDL-C), low-density lipoprotein cholesterol (LDL-C), and glycated hemoglobin A1C (HbA1C) were measured by commercial kits (Nanjing Jiancheng, Nanjing, China). The homeostasis model assessment of insulin resistance (HOMA-IR) formula was as follows: HOMA-IR = serum glucose levels (mmol/l) * insulin levels (mU/l) / 22.5. Serum TNFα, IL-6, IL-1β, and IL-10 levels were quantified using TNFα, IL-6, IL-1β, and IL-10 enzyme-linked immunosorbent assay (ELISA) kit (CUSABIO, Wuhan, China) following the manufacturer's instructions. Plasma LPS concentration was measured based on the Limulus Amebocyte lysate (LAL) kinetic chromogenic methodology (Macklin, T924484-32T) that measures color intensity directly related to the endotoxin concentration in a sample, follow by the previously reported[32].

**Insulin tolerance test and oral glucose tolerance test**. ITT was performed by injecting insulin (0.8 U/kg) intraperitoneally after a 12 h fasting. OGTT was performed by oral administration of ᴅ-glucose (2 g/kg) after overnight fasting. The level of blood glucose was measured using a glucose meter (AccuCheck, Roche, Switzerland) before insulin or glucose load (0 min) and at 15, 30, 60, 90, and

120 min after glucose load, and at 30, 60, 90, and 120 min after insulin load. The area of the curve (AOC) generated from the data collected during the OGTT were calculated by GraphPad 7.0.

**Histopathological examination**. Samples of liver and WAT were resected and fixed with 10% formaldehyde phosphate buffered saline (pH 7.4), then embedded in paraffin, sectioned, stained with hematoxylin/eosin (H&E stain), Oil Red O, and finally analyzed by light microscopy (Nikon Eclipse E200, Nikon, Japan).

**Real-time qPCR analysis**. Total RNA was extracted from the intestinal tissue with the TRIzol reagent according to the manufacturer's protocol (Invitrogen, Carlsbad, CA, USA), Quantification analysis of total RNA were performed by running 1 μL of sample on an Agilent 2100 bioanalyzer (Agilent RNA 6000 Nano kit, Agilent). Reverse transcription was performed on 2 μg of total RNA using a cloned AMV first-strand cDNA synthesis kit (Tiangen, Beijing, China). Primers used for cDNA amplification by real-time PCR are listed in Supplementary Table 2.

Glyceraldehyde-3-phosphate dehydrogenase (GAPDH) was used as the housekeeping gene for normalization of the target genes expression. PCR reactions were performed using KAPA SYBR FAST Qpcr Master Mix 2* kit (KK4601-07959389001) using a QuantStudio 6 Flex Real-Time PCR Systems.

**Western blotting**. Colon protein was extracted using RIPA lysis buffer (G2002; Servicebio) and PMSF (G2008; Beyotime). The protein amount was measured using a BCA protein assay reagent (P0012; Beyotime). Equal amounts of proteins were boiled in SDS-PAGE sample loading buffer (P0015L; Beyotime) for 15 min and then separated on a 10% SDS-PAGE gel and transferred onto a PVDF membrane (IPVH00010; Merck Millipore Ltd.). Membranes were soaked with 5% nonfat milk in TBST (20 mM Tris·Cl, 150 mM NaCl, and 0.05% Tween-20, pH 7.4) for 1 h at room temperature. Subsequently, membranes were incubated with primary antibodies at 4 °C overnight, followed by incubation with Alexa Fluor 790 goat anti-rabbit IgG H&L (1:10,000; ab186697; Abcam, Cambridge, UK) or Alexa Fluor 680 goat anti-mouse IgG (1:10,000; ab186694; Abcam). Finally, fluorescent signals were collected using an Odyssey infrared imaging system (LI-COR, Lincoln, NE). The intensities of protein bands were quantified with the Photoshop software (Adobe Systems, San Jose, CA), and the values were normalized to β-tubulin. Primary antibodies against DECTIN-1 (GB111816; 1:500) were purchased from Servicebio.

**Human serum and fecal samples**. Twenty-six T2DM patients and fifteen healthy individuals without chronic disease were included in our experiment, and we obtained blood from the vein for the β-glucan assay. Stool samples from 11 T2DM patients and 6 healthy individuals without chronic disease were collected (Supplementary Table 3), and DNA was extracted from fecal samples for the fungal microbiome. The study was approved by the Ethics Committee of Beijing Shijitan Hospital Affiliated to Capital Medical University (2017-035) and registered on the Chinese Clinical Trial Registry (ChiCTR2100042049), as well as conducted under the guidelines of the Helsinki Declaration.

**Microbial community analysis**. *Fungi*: the fungal microbiome was analyzed using high throughput sequencing technology. Over 83,234 paired-reads were collected from ITS2 regions using Illumina Novaseq platform. Of these, 65,181 reads with high quality sequences were selected for the later analysis. The reads were assigned to operational taxonomic units (OTUs) with a 97% similarity threshold and taxonomy assignment of the resulting OTU was carried out using the BLAST against the UNITE reference database.

The total fungi and the total *C. albicans* were assayed by a two-step nested PCR method for amplification as previously described[23]. Firstly, ITS fragments were amplified with the primers ITS1 and ITS4 using a KAPA HiFi HosStart Ready Mix (Kapa Biosciences, Wimington, MA), then the amplicon was performed for 2nd PCR using the primers 1737F and 2043R (Supplementary Table 2). Each genomic DNA sample (1 μL) was used for a 25 μl PCR mixture under the following conditions: 95°C for 5 min, 35 cycles of 45 s at 95 °C, 55 °C for 45 s, and 72 for 1 min, followed by 7 min at 72 °C.

**Culture of fungi**. Gut fungi in human fecal sample were cultured using a described method[38]. 100 mg of fecal samples from 6 healthy individuals and 11 T2D patients were diluted with 900 μL of phosphate buffer saline (PBS) for solid fungal culture. 50 μL 10-fold dilutions was used for culturing on Dixon agar (DIX) and PDA solid culture media supplemented with 3 antibiotics; namely, colistin (30 mg/ L), vancomycin (30 mg/ L) and imipenem (30 mg/ L) (Supplementary Table 1).

*C. albicans* strains were grown in antibiotics and incubated at 30 °C with shaking at 150 rpm for 24 h. Cell density was analyzed by measuring optical density at 600 nm in a microplate reader (Molecular Devices). Cell from YPD broth (Supplementary Table 1) were collected by centrifugation, washed twice with PBS and adjusted to the required cell density.

*Molecular fungal identification*. Direct ITS analysis was performed for fungal species isolates. The ITS1 (TCCGTAGGTGAACCTGCGG) and ITS4 (TCCTCCGCTTATTGATATGC) primers were used for PCR and sequencing. The fungi were identified on the basis of morphology and the DNA sequences of the ITS regions of their ribosomal RNA gene. Analysis of sequences was perfomed using BLAST in GenBank, and the results were seen in Supplementary Table 4.

*Preparation and analysis of 1,3-β-glucan isolated from* C. albicans. 1,3-β-glucan were isolated from the above mentioned *C. albicans* as previous reported (CAG) with modification[39]. Briefly, the cell wall was collected by centrifugation after 30 min ultrasonic fragmentation of *C. albicans*. After homogenization, extract with 1 M NaOH, then extract with 0.5 M acetic acid, and then wash with distilled water. Finally, the ether was used to dehydrate and dried at room temperature. The levels of CAG was measured with 1,3-β-DG ELISA kit according to manufacturer's instructions. The structural characteristics of CAG were analysis by HPGPC, IR and NMR. IR data was obtained on a Nicolet IS5FTIR spectrophotometer using the KBr-disk method. NMR spectra were acquired with Bruker Avance-500 spectrometer and CAG was dissolved in DMSO-$d_6$. HPLC using pre-column derivatization with 1-phenyl-3-methyl-5-pyrazolone (PMP) was used to analyze monosaccharides as previous described with minor revise[22]. In brief, 5.0 mg of GAG sample was mixed with 4 mL of 2 M trifluoroacetic acid (TFA) and then stirred at 120 °C for 2 h. After removal of TFA under vacuum, the hydrolyzed products CAG were mixed with 0.5 M PMP. The obtained PMP derivatives were analyzed by HPLC on an Agilent 1200 HPLC system equipped with a C-18 column (4.6 × 250 mm, 5 μm, Agilent) and eluted with a mixture of 0.1 M phosphate buffered saline/acetonitrile (15/85) at a flow rate of 1.0 mL/min. The injection volume was 20 μL and the UV detector was set at 245 nm.

**Statistical and reproducibility**. All assays were performed using $n = 8–9$ mice. All results are expressed as mean ± SEM. For multiple comparisons, statistical analysis was performed using the unpaired two-tailed Student's *t* test for single variables and two way ANOVA followed by the Tukey's multiple comparison tests with GraphPad 6.0.

**Reporting summary**. Further information on research design is available in the Nature Research Reporting Summary linked to this article.

## Data availability

The ITS2 sequencing data were deposited in the National Microbiology Data Center (Accession numbers: NMDC10018246). Source data for the graphs and charts are provided as Supplementary Data 1–5. Uncropped blot images were included as Supplementary Figure 3. Correspondence and requests for materials should be addressed to D. Yan.

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

## Acknowledgements

This work was supported by the National Natural Science Foundation of China (grant number 82073741 and 82130112), Youth Fund of Beijing Shijitan Hospital (2021-q07) and Youth Beijing Scholar (2022-051).

## Author contributions

Conceptualization, L.B. and D.Y.; Methodology, L.B., D.C.J., and G.Y.Z.; Investigation, L. B. and Y.Z.; Writing, L.B. and D.Y; Supervision, D.Y; Funding acquisition, L.B. and D.Y.

## Competing interests

The authors declare no competing interests.
