## [Peer Review File · Communications Biology]

Reviewers' comments:

Reviewer #1 (Remarks to the Author):

In this study, the authors evaluated whether gut mycobiota play a role in the occurrence and progress of T2D and related metabolic disorders by treated HFD-fed mice with or without the antifungal agent amphotericin B. They found that elimination of gut mycobiota alleviated insulin resistance and related metabolic disorders in HFD-fed mice. By analyzing stool samples from patients with type 2 diabetes and healthy volunteers, the authors found that gut mycobiota alterations in T2D patients. They discovered a significant enrichment of *C. albicans* in the T2D group and thought that *C. albicans* as a potential pathogen in the pathological process of insulin resistance and diabetes. After repopulated HFD-mice with *C. albicans*, the authors found that *C. albicans* significantly accelerated the features of T2D and insulin resistance in the HFD-fed mice. Moreover, they found that the β -glucan from *C. albicans* mirrored the deteriorating effect of *C. albicans* through the dectin-1 dependent pathway. The study was interesting. However, several concerns need to be addressed.

1. Was there any difference between gut mycobiota in mice fed with normal diet and HF diet?
2. In result part '*C. albicans* accelerated the progress of T2D and related disorders', the normal diet mice were also needed to treat with or without *C. albicans*.
3. In result part '*CAG* mirrored the pathogenic effect of *C. albicans*', again, the normal diet mice were needed to treat with or without *CAG*.

Reviewer #2 (Remarks to the Author):

In the following manuscript, the authors detail a nice progressive mechanism demonstrating the impact of the gut mycobiome on glucose homeostasis. The authors detail an effect of the mycobiome in general, then a specific fungus (CA), and then a 'metabolite' of that fungus (beta-glucan). However, there are several major concerns with the manuscript:

- 1) The metabolic assessments do not follow standard methods used in all metabolic disease studies. First, the authors collect blood at 40,80,120min after OGTT and ITT - this is not acceptable. For OGTT it must be 15,30,60,90,120 - and a postprandial insulin measurement at 15min. The data is extremely difficult to interpret without that format. For ITT it should be 30,60,90,120. Second, ITT data should be analyzed as % from baseline glucose. This is extremely important in Figure 1, as it looks like there is actually no difference in response to insulin between HFD and HFD+antifungal - it is simply due to the fact they start at a higher BG. The authors state in the text - 'a profound impact on insulin resistance' - with that data I would argue there is no impact at all on insulin resistance... also, in mice, glucose clamps and not ITTs are the gold standard for truly measuring insulin sensitivity. Third, OGTTs are done after overnight fast, a 4-6hr fast is the standard for mice. These need to be done properly to interpret the data correctly.
- 2) While there is an effect with the anti-fungal, there are several concerns. First, are they sure AmB did eliminate the fungi? There is no proof of this. Second, how do they know it did not affect other physiology? To be sure, they should perform in GF mice and make sure AmB does not have any direct effects on the host. Along these lines, how do the authors know that AmB does not just cause aversion and reduced food intake that then drives all the improvements. Considering there was a decrease in body weight, it might be caused by an aversion to the treatment. This must be addressed, and a pair-fed group must be utilized if you want to claim the fungi are selectively improving glucose homeostasis, otherwise the changes in OGTT might simply be due to reduced body weight. Third, does it also impact bacteria which could be the reason for the effect? The authors need to do sequencing of the mice in Figure 1. Fourth, the authors show a drastic change in fasting glucose on Fig 1A but then in the OGTT there is no difference in fasting glucose - why?
- 3) For both the fungi CA and the beta-glucan - what is the rationale for the doses used? What does treating with those doses actually do to the endogenous levels - are they comparable to what they

observe with just HFD-feeding? Also, why were they not given in chow-fed mice as well?

4) Some of the data do not link with each other - mainly the fact that CA treatment impairs glucose homeostasis with no change in BW but then directly giving beta-glucan does increase BW. The authors need to address this discrepancy.

5) It is unclear why the authors look at changes to the gut microbiome - they show the effect of beta-glucan is completely restored with the dectin-1 antagonist.

6) There is one key experiment missing with the link of CA-beta-glucan-dectin-1 and that is CA treatment with dectin-1 antagonist. That would fully link the mechanism, and without it, you can only speculate that CA is worsening glucose through the beta-glucan-dectin-1 pathway.

7) Grammar and writing needs to be greatly improved. It is difficult to read at times, and much of the terminology is not properly used (like free-feeding?)

Reviewer #3 (Remarks to the Author):

In the present study, authors have investigated in mice and in humans the importance of the mycobiota (gut fungi) in the context of obesity, type 2 diabetes, and other related metabolic disorders. They found out that depletion of the commensal fungi in mice can protect them from the effect induced by high fat (HF) feeding. Repopulation with *C. albicans* in HF diet mice accelerated insulin resistance and related disorders, and β -glucan from *C. albicans* could reproduce the same deteriorating effect of *C. albicans*. I really appreciate the huge work performed by the authors, and the novelty of the study proving the importance of fungi in the context of metabolic disorders. However, there are some comments that must be addressed, please see below:

Major comments:

The HFD 60% is commonly used to identify the several mechanisms associated with the etiology of obesity and related metabolic disorders, but I am wondering about the CT diet used for this study. It was not mentioned in the M&M section.

In order to have an idea of the variability among mice of the same group, bar plots have to be replaced with a scatter dot plot. This will help the reader to visualize the number of mice (N) per group, which I found to be quite confusing from one study to another one, and it is not always the same. I highly recommend clarifying this point.

Based on which analysis mice were excluded?

Minor comments:

Introduction and Abstract

Line 10: For the GUDCA please refer to the full name, it is the first time that has been mentioned in the text.

Line 26: Same comment for ITS2

Results

Eliminate of gut mycobiota alleviated insulin resistance and related metabolic disorders in HFD-fed mice

Line 10: Please rephrase this sentence. I would say depletion of the gut mycobiota by Amb....

Please check Graphs C and D: align symbols (# and * correctly). Moreover, symbol # is not reported in the legend of the figures and to which group it refers.

Legend Fig.1: A space between one and way is missing. Same comment for all the figures legends.

Gut mycobiota alternations in T2D patients

Lines 16 and 21: The f in the Fig must be capital as with the rest.

Please improve the quality of Fig.2, it is quite difficult to read out most of the names, and for Fig.2I the image is overlapping with the Y axis.

C. albicans accelerated the progress of T2D and related disorders.

Figure 3, Line 11: N=5? Based on what authors have excluded these mice from the analysis? In the next line, they show to have N: 8-10. Please clarify it! Moreover, the scale bars is missing for all the images of Fig 3I.

CAG mirrored the pathogenic effect of C. albicans

Figure 5: Please check the sizes of the graphs and some Y axes are cut.

CAG induce insulin resistance though the activation of dectin-1 pathway

Line 22: Replace Ocludin with Occludin.

Figure 6, Line 6: N= 5-8?

Figure 7: N= 10?

The bacterial community change induced by C. albicans and CAG

Please improve the quality of Fig.8, some words and names are difficult to read out.

Discussions

Line 11: Replace eliminate with eliminating

Line 24: sapmles must be replaced with samples

Materials and Methods

The number of mice (N) is not properly reported for each animal experiment. Please clarify it.

The recommends from editorial office has changed as suggested and double checked carefully point by point.

Reviewers' comments:

Reviewer #1 (Remarks to the Author):

In this study, the authors evaluated whether gut mycobiota play a role in the occurrence and progress of T2D and related metabolic disorders by treated HFD-fed mice with or without the antifungal agent amphotericin B. They found that eliminate of gut mycobiota alleviated insulin resistance and related metabolic disorders in HFD-fed mice. By analyzing stool samples from patients with type 2 diabetes and healthy volunteers, the authors found that gut mycobiota alternations in T2D patients. They discovered a significant enrichment of *C. albicans* in the T2D group and thought that *C. albicans* as a potential pathogens in the pathological process of insulin resistance and diabetes. After repopulated HFD-mice with *C. albicans*, the authors found that *C. albicans* significantly accelerated the features of T2D and insulin resistance in the HFD-fed mice. Moreover, they found that the β -glucan from *C. albicans* mirrored the deteriorating effect of *C. albicans* through the dectin-1 dependent pathway. The study was interesting. However, several concerns need to be addressed.

1. Was there any difference between gut mycobiota in mice fed with normal diet and HF diet?

ITS-2 sequencing analysis was used to evaluate the fungal changes between chow diet and HFD mice (**Response figure 1A-C**), and we observed that high fat diet can also induce the enrichment of *Candida* fungi (**Response figure 1D**), which is consistent with those in T2D population.

Response figure 1. Gut mycobiota change between CD and HFD.

(A) α -diversity of the gut mycobiota between mice fed with CD and HFD, as indicated by the ACE and Shannon indices. (B) Partial least squares discriminant analysis (PLS-DA) using the Bray-Curtis distance. (C) Fungal taxonomic profiling of the gut mycobiota from different samples at the genus level.

2. In result part '*C. albicans* accelerated the progress of T2D and related disorders', the normal diet mice were also needed to treat with or without *C. albicans*.

We performed the experiment that the chow-diet mice were treated with *C. albicans*, and we observed that *C. Albicans* did not induce metabolic disorder in mice, suggesting that *C. albicans* can not affect metabolic phenotypes in health individual.

3. In result part '*CAG* mirrored the pathogenic effect of *C. albicans*', again, the normal diet mice were needed to treat with or without *CAG*.

We performed the experiment that the chow-diet mice were treated with *CAG*, and we observed that *CAG* did not induce metabolic disorder in mice, suggesting that *CAG* can not affect metabolic phenotypes in health individual, which is similar to that of *C. albicans* treated mice.

Reviewer #2 (Remarks to the Author):

In the following manuscript, the authors detail a nice progressive mechanism demonstrating the impact of the gut mycobiome on glucose homeostasis. The authors detail an effect of the mycobiome in general, then a specific fungus (CA), and then a 'metabolite' of that fungus (beta-glucan). However, there are several major concerns with the manuscript:

1) The metabolic assessments do not follow standard methods used in all metabolic disease studies. First, the authors collect blood at 40,80,120min after OGTT and ITT - this is not acceptable. For OGTT it must be 15,30,60,90,120 - and a postprandial insulin measurement at 15min. The data is extremely difficult to interpret without that format. For ITT it should be 30,60,90,120. Second, ITT data should be analyzed as % from baseline glucose. This is extremely important in Figure 1, as it looks like there is actually no difference in response to insulin between HFD and HFD+antifungal - it is simply due to the fact they start at a higher BG. The authors state in the text - 'a profound impact on insulin resistance' - with that data I would argue there is no impact at all on insulin resistance... also, in mice, glucose clamps and not ITTs are the gold standard for truly measuring insulin sensitivity. Third, OGTTs are done after overnight fast, a 4-6hr fast is the standard for mice.

These need to be done properly to interpret the data correctly.

We re-performed standardized tests on OGTT and ITT according to the reviewer's requirements; We agree with the reviewers that glucose clamps is the gold standard for detecting insulin resistance, but we cannot detect it due to the current condition. Accordingly, HOMA-IR is another important index for evaluating insulin resistance in mice, and we performed it on all models (PMID: 31332392, 30397356, 34329568).

2) While there is an effect with the anti-fungal, there are several concerns.

First, are they sure AmB did eliminate the fungi? There is no proof of this.

We used qPCR to detect the eliminating effect of AmB on fungi (**Figure S1**)

Second, how do they know it did not affect other physiology? To be sure, they should perform in GF mice and make sure AmB does not have any direct effects on the host. Along these lines, how do the authors know that AmB does not just cause aversion and reduced food intake that then drives all the improvements. Considering there was a decrease in body weight, it might be caused by an aversion to the treatment. This must be addressed, and a pair-fed group must be utilized if you want to claim the fungi are selectively improving glucose homeostasis, otherwise the changes in OGTT might simply be due to reduced body weight.

We agree with the reviewers that the use of AmB may cause potential side effects. However, as the comparison between sterile mice and SPF mice will lead to more bacterial interference, there is still no appropriate method to accurately separate bacteria and fungi in the intestinal tract for colonization. At the same time, because AMB is not absorbed in the intestine, its role is basically limited to the intestine, so it is widely used in the functional study of intestinal fungi. In order to explain that the effect is caused by the elimination of fungi rather than the effect of AmB itself, we added two antifungal drugs, fluconazole and 5-fluorouracil, which are also commonly used in the functional verification of intestinal fungi (**Figure 1**). The above three antifungal drugs can effectively kill intestinal fungi and significantly improve the metabolic disorder induced by DIO. Finally, the three antifungal drugs had no significant effect on the food intake of mice, suggesting that the decrease in appetite may not be caused by aversion.

Third, does it also impact bacteria which could be the reason for the effect? The authors need to do sequencing of the mice in Figure 1.

We analyzed the bacterial composition of mice by 16S RNA sequencing analyses. AmB treated mice had no significant effect on bacterial composition (**Response figure 2**), suggesting that bacteria may not be the direct effect of AmB on improving metabolic disorders.

Response figure 2. Gut bacteria change after antifungal agents treatment..

(A) α -diversity of the gut bacteria between mice treated with different antifungal agents, as indicated by the ACE and Shannon indices. (B) Partial least squares discriminant analysis

(PLS-DA) using the Bray-Curtis distance. (C) Fungal taxonomic profiling of the gut bacteria from different samples at the genus level.

Fourth, the authors show a drastic change in fasting glucose on Fig 1A but then in the OGTT there is no difference in fasting glucose - why?

Fasting blood glucose was fasting for 8 hours, while OGTT was fasting for 16 hours, so there was a difference.

3) For both the fungi CA and the beta-glucan - what is the rationale for the doses used? What does treating with those doses actually do to the endogenous levels - are they comparable to what they observe with just HFD-feeding? Also, why were they not given in chow-fed mice as well?

The treatment of *C. albicans* and CAG were referred to the previous study, and is consistent with the content in mice under HFD conditions; We also evaluated the effects of *C. albicans* and CAG treatment on metabolic disorders under chow diet conditions according to the reviewers' recommendations (**Figure 3 and 5**).

4) Some of the data do not link with each other - mainly the fact that CA treatment impairs glucose homeostasis with no change in BW but then directly giving beta-glucan does increase BW. The authors need to address this discrepancy.

As an important intestinal symbiotic fungus, the mechanism of *C. albicans* is complex. Recent studies have found its potential protective effect on Th17 cells (PMID: 35296857), so there is no effect on body weight may be caused by complex effects. We also proved that CAG mainly affects body weight and blood glucose disorder through Dectin-1 pathway by using Dectin-1 inhibitor.

5) It is unclear why the authors look at changes to the gut microbiome - they show the effect of beta-glucan is completely restored with the dectin-1 antagonist.

The interaction between fungi and bacteria is an important issue in the study of intestinal fungi, so we observed the effects on bacteria; As the reviewer said, the Dectin-1 inhibitor completely antagonized the effect of CAG, suggesting that bacteria may have little effect on the effect of CAG. In order not to confuse readers, we removed the sequencing results of bacteria in this part in the revised version.

6) There is one key experiment missing with the link of CA-beta-glucan-dectin-1 and that is CA treatment with dectin-1 antagonist. That would fully link the mechanism, and without it, you can only speculate that CA is worsening glucose through the beta-glucan-dectin-1 pathway.

We performed *C. albicans* treatment with dectin-1 antagonist experiment according to the suggestion of reviewer. The results showed that after Dectin-1 treatment, the effect of *C. albicans* on glucose metabolism disorder was alleviated, which proved that *C. albicans* affect metabolism disorder through beta-glucan-dectin-1 pathway (**Figure 7**).

7) Grammar and writing needs to be greatly improved. It is difficult to read at times, and much of the terminology is not properly used (like free-feeding?)

We revised the grammar and writing in manuscript according to the reviewers' suggestions.

Reviewer #3 (Remarks to the Author):

In the present study, authors have investigated in mice and in humans the importance of the mycobiota (gut fungi) in the context of obesity, type 2 diabetes, and other related metabolic disorders. They found out that depletion of the commensal fungi in mice can protect them from the effect induced by high fat (HF) feeding. Repopulation with *C. albicans* in HF diet mice accelerated insulin resistance and related disorders, and β -glucan from *C. albicans* could reproduce the same deteriorating effect of *C. albicans*. I really appreciate the huge work performed by the authors, and the novelty of the study proving the importance of fungi in the context of metabolic disorders. However, there are some comments that must be addressed, please see below:

Major comments:

The HFD 60% is commonly used to identify the several mechanisms associated with the etiology of obesity and related metabolic disorders, but I am wondering about the CT diet used for this study. It was not mentioned in the M&M section.

We re-evaluated the fungal changes under chow-diet, and conducted *C. albicans* and CAG treatment under chow-diet. The results showed that *C. albicans* and CAG could not cause metabolic disorder in mice, suggesting that *C. albicans* and CAG were not enough to affect metabolic phenotypes under normal conditions.

In order to have an idea of the variability among mice of the same group, bar plots have to be replaced with a scatter dot plot. This will help the reader to visualize the number of mice (N) per group, which I found to be quite confusing from one study to another one, and it is not always the same. I highly recommend clarifying this point. Based on which analysis mice were excluded?

We replaced the histogram with a scatter chart and redefined the animals numbers in different experiments.

Minor comments:

Introduction and Abstract

Line 10: For the GUDCA please refer to the full name, it is the first time that has been mentioned in the text.

We revised it according to the reviewers' suggestions.

Line 26: Same comment for ITS2

Results

Eliminate of gut mycobiota alleviated insulin resistance and related metabolic disorders in HFD-fed mice

Line 10: Please rephrase this sentence. I would say depletion of the gut mycobiota by Amb....

We revised it according to the reviewers' suggestions.

Please check Graphs C and D: align symbols (# and * correctly). Moreover, symbol # is not reported in the legend of the figures and to which group it refers.

We revised the figure legend according to the reviewers' suggestions.

Legend Fig.1: A space between one and way is missing. Same comment for all the figures legends.

We revised the figure legend according to the reviewers' suggestions.

Gut mycobiota alternations in T2D patients

Lines 16 and 21: The f in the Fig must be capital as with the rest.

We revised them according to the reviewers' suggestions.

Please improve the quality of Fig.2, it is quite difficult to read out most of the names, and for Fig.2I the image is overlapping with the Y axis.

We adjusted Fig 2 according to the reviewers' suggestions.

C. albicans accelerated the progress of T2D and related disorders.

Figure 3, Line 11: N=5? Based on what authors have excluded these mice from the analysis? In the next line, they show to have N: 8-10. Please clarify it! Moreover, the scale bars is missing for all the images of Fig 3I.

We revised them according to the reviewers' suggestions.

CAG mirrored the pathogenic effect of C. albicans

Figure 5: Please check the sizes of the graphs and some Y axes are cut.

We adjusted the pictures according to the reviewers' suggestions.

CAG induce insulin resistance though the activation of dectin-1 pathway

Line 22: Replace Ocludin with Occludin.

We have made corrections.

Figure 6, Line 6: N= 5-8?

We marked all N value.

Figure 7: N= 10?

We marked all N value.

The bacterial community change induced by C. albicans and CAG

Please improve the quality of Fig.8, some words and names are difficult to read out.

We adjusted the pictures according to the reviewers' suggestions.

Discussions

Line 11: Replace eliminate with eliminating

We have made corrections.

Line 24: sapmles must be replaced with samples

This is a clerical error. We have made corrections.

Materials and Methods

The number of mice (N) is not properly reported for each animal experiment. Please clarify it.

We marked all N value.

Reviewers' comments:

Reviewer #1 (Remarks to the Author):

The text modifications and added data improved the readability and reliability of the results.

Reviewer #2 (Remarks to the Author):

The authors have addressed all the comments and have greatly improved the manuscript.

Reviewer #3 (Remarks to the Author):

In the following manuscript the authors have investigated the important role of the gut mycobiota in the progression of type 2 diabetes (T2D) and related metabolic disorders in mice. They also have shown that the number of fungi is significantly higher in T2D patients when compared to the healthy group. The major novelty of this study is to show the importance of fungi in the onset of T2D and related metabolic disorders (e.g., steatosis, inflammation). However, there are key limitations of the following manuscript that need to be addressed:

(i) To prove the importance of fungi in the onset of such metabolic disorders CT mice should have been treated in the same manner as the HF-fed mice. These changes might be more diet-related since it is well-known in the literature that HF feeding is associated with drastic changes in the composition of the gut microbiota. Therefore, authors cannot really dissect the link between fungi and bacteria due to the presence of a dysbiosis condition.

(ii) The OGTT test is not properly performed, and therefore authors cannot really claim all the effect associated with the glucose metabolism (this is a key limitation of the study).

General comment:

Please carefully check the full manuscript, there are many typing mistakes.

Other comments:

Which kit was used to measure the plasma LPS levels?

The M&M section for the OGTT test does not really reflect what the authors have done (this is a key limitation of the study, and authors cannot claim the beneficial effect on glucose metabolism).

Lack of depository data for the gut mycobiota analysis

The recommends from editorial office has changed as suggested and double checked carefully point by point.

Reviewers' comments:

Reviewer #3 (Remarks to the Author):

In the following manuscript the authors have investigated the important role of the gut mycobiota in the progression of type 2 diabetes (T2D) and related metabolic disorders in mice. They also have shown that the number of fungi is significantly higher in T2D patients when compared to the healthy group. The major novelty of this study is to show the importance of fungi in the onset of T2D and related metabolic disorders (e.g., steatosis, inflammation). However, there are key limitations of the following manuscript that that need to be addressed:

(i) To prove the importance of fungi in the onset of such metabolic disorders CT mice should have been treated in the same manner as the HF-fed mice. These changes might be more diet-related since it is well-known in the literature that HF feeding is associated with drastic changes in the composition of the gut microbiota. Therefore, authors cannot really dissect the link between fungi and bacteria due to the presence of a dysbiosis condition.

We agree with the reviewer that “HF feeding is associated with drastic changes in the composition of the gut microbiota”, and we conducted *C. albicans* and CAG treatment under chow-diet (CT mice) and found that *C. albicans* and CAG could not cause metabolic disorder in chow-diet fed mice, suggesting that *C. albicans* and CAG were not enough to affect metabolic phenotypes under normal conditions. We added the condition of chow-diet treatment in the M&M section.

(ii) The OGTT test is not properly performed, and therefore authors cannot really claim all the effect associated with the glucose metabolism (this is a key limitation of the study).

We re-performed standardized tests on OGTT in the revision version according to the reviewer's requirements, which follow the standard recommendation (PMID: 34117483). For further standardization our results, we change AUC to a more rigorous area of the curve (AOC), and re-describe in the M&M section.

General comment:

Please carefully check the full manuscript, there are many typing mistakes.

We carefully corrected the descriptive errors in the article, as the reviewer suggested.

Other comments:

Which kit was used to measure the plasma LPS levels?

Plasma LPS concentration was measured based on the Limulus Amebocyte lysate (LAL) kinetic chromogenic methodology (Macklin, T924484-32T) that measures color intensity directly related to the endotoxin concentration in a sample, follow by the previously reported (PMID: 23671105) .

The M&M section for the OGTT test does not really reflect what the authors have done (this is a key limitation of the study, and authors cannot claim the beneficial effect on glucose metabolism).

We change AUC to a more rigorous area of the curve (AOC), and re-describe in the M&M section.

Lack of depository data for the gut mycobiota analysis

We deposit the gut mycobiota analysis data on a public database (China National Microbiology Data Center (NMDC), under accession numbers NMDC10018246).

REVIEWERS' COMMENTS:

Reviewer #3 (Remarks to the Author):

I really appreciate that the authors have addressed many of reviewer's comments and improved the manuscript. However, the manuscript has still important flaws and my additional comments will not bring any additional values at this stage of the review.

Dr. Sridhar Mani (Remarks to the Author):

The authors did a study using HFD + *C. albicans* versus HFD alone -- so this should not be an issue-- there other studies in the manuscript that also support *C. albicans* or its beta-glucan mirrors these findings. I think that comment should not be a sticking point for approval.

Final Revision Instructions

To the Author— Please review the editorial comments and requests below and confirm that changes have been made in the manuscript in the right-hand column. **This document must be uploaded as a related manuscript file.**

Please see our final file submission checklist for information about submitting your revised documents.

Files and General Policies	
Main manuscript file must be in Microsoft Word or LaTeX format. LaTeX and Tex article source files must be accompanied by the compiled PDF for reference. The bibliography must be submitted separately (as a .bib file) or contained within the .tex file.	Microsoft Word
Each Figure must be provided as a separate file and must be supplied whole, with all panels included in a single document. Figures should be provided at a minimum resolution of 300 dpi at final size. This applies to Supplementary Figures. Figure files must only contain images (please also leave out labels such as “Figure 1” etc). Figure captions must instead be included within the main manuscript file, grouped together at the end of the document.	Figures were be provided as a separate file with the size of 300 dpi.
All figures, tables, and supplementary items must be cited in the manuscript and numbered in the order in which they appear.	They are cited and numbered in the order.
Please check whether your manuscript contains third-party images, such as figures from the literature, stock photos, clip art or commercial satellite and map data. We strongly discourage the use or adaptation of previously published images, but if this is unavoidable, please request the necessary rights documentation to re-use such material from the relevant copyright	NO

holders and return this to us when you submit your revised manuscript. An appropriate permissions statement must be present in the relative figure caption for any third-party images.	
Please check that you have not copied any text directly from published work (even your own) without clear attribution, including one or more references. We run a plagiarism detection software and may need to request additional changes if we identify large blocks of identical text.	NO
An updated editorial policy checklist that verifies compliance with all required editorial policies must be completed and uploaded with the revised manuscript. All points on the policy checklist must be addressed; if needed, please revise your manuscript in response to these points. https://www.nature.com/documents/nr-editorial-policy-checklist.pdf. Please note that this form is a dynamic 'smart pdf' and must therefore be downloaded and completed in Adobe Reader. This file will not open in an internet browser.	Yes
The reporting summary will be published alongside your manuscript therefore it needs to accurately represent your work. In this case, please take a closer look at the reporting summary and make sure things are completed correctly. If an item does not apply, for example human participants, I need you to check the NA box next to that item. No section should be left blank. Also, please make sure to include your name and date at the top of the document. If you require a new Reporting Summary form, please download it here: https://www.nature.com/documents/nr-reporting-summary.pdf. Please note that this form is a dynamic 'smart pdf' and must therefore be	Yes

downloaded and completed in Adobe Reader. This file will not open in an internet browser.	
Your paper will be accompanied by a brief editor's summary when it is published on our homepage. Please approve the draft summary below or provide us with a suitably edited version (no more than 250 characters including spaces). β-glucan from Candida albicans promotes high-fat diet-induced insulin resistance and related metabolic disorders in mice, and C. albicans is enriched in type 2 diabetes (T2D) patients, supporting a role of gut mycobiota in the progress of T2D.	The drafts summary provided by the editor is OK.
ORCID Communications Biology is committed to improving transparency in authorship. As part of our efforts in this direction, we are now requesting that all authors identified as ‘corresponding author’ create and link their Open Researcher and Contributor Identifier (ORCID) with their account on the Manuscript Tracking System (MTS) prior to acceptance. ORCID helps the scientific community achieve unambiguous attribution of all scholarly contributions. For more information please visit http://www.springernature.com/orcid. For all corresponding authors listed on the manuscript, please follow the instructions in the link below to link your ORCID to your account on our MTS before submitting the final version of the manuscript. If you do not yet have an ORCID you will be able to create one in minutes. https://www.springernature.com/gp/researchers/orcid/orcid-for-nature-research	Prof. Dan Yan ORCID No.: 0000-0003-2985-3798 Dr. Li Bao ORCID No.: 0000-0002-2879-3490

IMPORTANT: All authors identified as ‘corresponding author’ on the manuscript must follow these instructions. Non-corresponding authors do not have to link their ORCIDs but are encouraged to do so. Please note that it will not be possible to add/modify ORCIDs at proof. Thus, if they wish to have their ORCID added to the paper they must also follow the above procedure prior to acceptance. To support ORCID's aims, we only allow a single ORCID identifier to be attached to one account. If you have any issues attaching an ORCID identifier to your MTS account, please contact the Platform Support Helpdesk at http://platformsupport.nature.com/	
We regularly highlight papers published in Communications Biology on the journal’s Twitter account (@CommsBio). If you would like us to mention authors, institutions, or lab groups in these tweets, please provide the relevant twitter handles in the right-hand column.	No
We would welcome the submission of material for the ‘Featured Image’ section on the Communications Biology home page. Images should relate to the content of your manuscript but need not be contained within the paper. Photographs and aesthetically interesting images are preferred; diagrams are generally not used. Suggestions should be uploaded as a Related Manuscript file. Please provide 1200x675-pixel RGB images. You will also need to submit a completed Image License to Publish. Unfortunately, we cannot promise that your suggestions will be used.	There is no featured image.
Supplementary information	
Supplementary Information Format and referencing  ● Supplementary Figures, small Tables, and any supplementary text 	Supplementary Information including figures and tables is provided in a single pdf, and we revised the order of supplementary tables.

must be provided in a single PDF. Figures and their captions should be presented together.  ○ If you include a title page, please check that the title and author list matches the main manuscript. ● All Supplementary items must be referred to in the manuscript, and items must be mentioned in numerical order. Please do not include general references to “Supplementary Material”; instead refer to specific items. ● Additional files can be provided as Supplementary Data (Excel files, text files, .zip folders), Supplementary Movies, Supplementary Audio, or Supplementary Software (.zip folder) Supplementary Information files will be uploaded with the published article as they are submitted with the final version of your manuscript. Any highlighting or tracked changes should be removed from the file. Please cite supplementary tables in order.	
Supplementary items must be cited in a consistent format. Names of items in the Supplementary file(s) must match those used in the main manuscript. We recommend using the following naming formats: Supplementary Figure 1, Supplementary Table 1, Supplementary Data 1, Supplementary Note 1, and Supplementary References.	Names of tables and figures in the Supplementary file(s) have been revised as Supplementary Figure 1-3, Supplementary Table 1-4.
It’s mandatory to provide access to the numerical source data (raw data) for all graphs and charts in the main and supplementary figures: We strongly recommend depositing these to suitable repositories (such as Figshare, Dryad, or a data type-specific repository if one exists). Otherwise, all source data underlying the graphs and charts presented in the main figures must be uploaded as Supplementary Data (in Excel or text	We have upload all source data presented in the main figures as Supplementary Data 1 (in Excel format) and cite in the Data Availability statement

format). Note that only the data used directly for generating the charts needs to be supplied. Please provide the source data as Supplementary Data 1. Please cite it in the Data Availability statement.	
For any Supplementary Files such as those mentioned above that are not included your combined PDF (e.g. Supplementary Data, Movies, Audio, Software), please provide a title and description for each file here in the column to the right. For example: File name: Supplementary Data 1 Description: The source data behind the graphs in the paper	Supplementary Data 1 Description: The source data behind the figures in the paper
Title Page	
Please ensure that the author list provided in our manuscript tracking system matches the author list in the main manuscript.	Yes
Please check that your author list and affiliations comply with the following:  ● Affiliations must be numbered in the order of their first appearance in the author list. Please use numbers instead of letters for affiliations.	Yes
Manuscript title Please ensure the title clearly describes the central finding of the paper. We recommend writing the title as a declarative statement of approximately 15	It is OK, I have revised it as editor's suggestion.

words or fewer. Be sure to include any key species, protein names, or gene names to ensure optimal retrieval of the paper in database searches. The editors recommend the following title: Abnormal proliferation of gut mycobiota contributes to the aggravation of Type 2 diabetes	
Abstract The abstract should be accessible to non-specialists and avoid jargon and abbreviations. Please write the abstract in the present tense. We recommend structuring the abstract as follows: Opening statement explaining why this area of research is important. A sentence explaining the gap in knowledge that your research will address. Here we show (or an equivalent phrase), and then the major results and conclusions of the paper. Final sentence indicating any broader impacts and how this research will be used in the future. The editors recommend the following edits to your abstract: Please define HFD and ITS2.	We have revised it as editor’s suggestion.
Main text	
Format of the main text Please ensure your manuscript includes the following sections, presented in this order:  1. “Introduction”: The background and rationale for the work. The final paragraph should be a brief summary of the major results and conclusions. The results of the current study must only be discussed in 	We have checked the main text carefully, and it met the requirement.

this final paragraph. The Introduction should contain no references to figures or tables. Do not include subheadings. 2. “Results” or “Results and Discussion”. This should be split into subheaded sections; we recommend 1 subheading per main figure or table. Figures should not be embedded in the text but submitted separately. a. Do not use more than 1 layer of subheadings. b. A “Conclusions” paragraph can be included only if the results and discussion are combined into a single section. 3. “Discussion” (optional), without subheadings. 4. Methods, which should be split into subheaded sections. Do not use more than 1 layer of subheadings. To improve readability, we recommend that the main text (Introduction, Results and Discussion) be limited to approximately 5000 words or fewer. Please retitle the Methods section.	
Statistical reporting Wherever statistics have been derived (e.g. error bars, box plots, statistical significance) the legend needs to provide and define the n number (i.e. the sample size used to derive statistics) as a precise value (not a range), using the wording “n=X biologically independent samples/animals/independent experiments” etc. as applicable. Please provide and define the n number in Fig. 2a, 2i, 4a, S1, S3d, S3h.	We have revised it, and FigS3-4 has little to do with this article and has been deleted.
Please include exact p-values where possible. We ask that you also include the name of the statistical test and the estimated effect size. If applicable, please also include the confidence interval.	We have revised it.

Please include p-values and tests in 2a, 2i, S3d, S3h.	
We recommend editing the main text for English language and grammar to improve readability and clarity for our readers. If you would like the assistance of paid editing services to do this, we can recommend our affiliates, Nature Research Editing Service: https://authorservices.springernature.com/language-editing and American Journal Experts: https://www.aje.com/go/springernature Please note that use of an editing service is neither a requirement nor a guarantee of publication. Free assistance is available from our resources page: https://www.springernature.com/gp/researchers/campaigns/english-language-forauthors	We have checked the grammar carefully to clarify the manuscript.
Display items	
Figure captions/legends Figures must have a title that will appear above the Figure and a legend that will appear below the Figure (see e.g. https://www.nature.com/articles/s42003-020-1059-1/figures/1) The Figure title must describe the Figure as a whole and must not contain reference to specific figure panels. The Figure legend must refer to and describe all panels. Abbreviations, symbols, colors, and shading present in the Figure must be defined. Please write out the symbols/colors in words (blue circles, red dashed line, etc.) within these definitions. All figure panels must be labelled using lower case letters. Please refrain from referring to sections of figures as top/bottom/left/right/, etc.	We have revised it.

Please use lower case letters to label panels. Please define each line in Fig. 4e. Please label all panels in Fig. S4.	
Axis and panel labels will be published as received. We recommend using a sans-serif font such as Arial or Helvetica. Please be aware that some axis labels are too small, such as Fig. 2c.	We have revised it.
Data presentation in bar graphs and line graphs For all graphs depicting a single point value (e.g., mean) with error bars, you must add individual data points or convert the graph to a boxplot or dot-plot. You may wish to refer to this blog post about representing data distribution in plots (particularly for small datasets). We strongly encourage the same for plots with multiple time courses depicted. See the June 24, 2019 CommsBio editorial for more details about this policy. Example plots are shown here:	We have revised it.

Examples of plots showing data distribution. Figure 2 from the editorial linked to above.

Please provide individual data points for Fig. 2a, 2i, 4a, 6, S1, S3d, S3h.

When choosing a color scheme please consider how it will display in black and white (if printed), and to users with color blindness. Please consider distinguishing data series using line patterns rather than colors, or using optimized color palettes such as those found at <https://www.nature.com/articles/nmeth.1618>.

The use of colored axes and labels should be avoided.

Please avoid the use of red/green color contrasts in Fig. 2h, 4e, S3g, S4f, as these may be difficult to interpret for colorblind readers.

We have revised it.

Please define the error bars in each Figure and Supplementary Figure where they are used. One statement at the end of each Figure caption is sufficient if the error bars are equivalent throughout the Figure. Please define error bars in 2a, 2i, S1, S3d, S3h.	
Microscopy images and photographs in each Figure and Supplementary Figure must be accompanied by scale bars, and these must be defined. Please add scale bars and corresponding definitions to Fig. 1o, 3i, 5i.	We have revised it in figure legends (Scale bars, 100µm).
Blots and gels All blots/gels must be accompanied by size markers in every figure panel. Uncropped and unedited blot/gel images must be included as Supplementary Figure(s). The new Supplementary Figure(s) must be cited in the main manuscript text (for example, in the Data Availability Statement). Please pay close attention to our Digital Image Integrity Guidelines and to the following points below:  ● that unprocessed scans are clearly labelled and match the gels and western blots presented in figures. Unprocessed scans must be included in a supplementary figure. ● that control panels for gels and western blots are appropriately described as loading on sample processing controls ● all images in the paper are checked for duplication of panels and for splicing of gel lanes. Finally, please ensure that you retain unprocessed data and metadata files after publication, ideally archiving data in perpetuity, as these may be requested during the peer review and production process or after publication if any issues arise.	Uncropped blot images were included as Supplementary Figure 3

Please add size markers in Fig. 6a, 6b. Please also provide uncropped images as Fig. S5. Please cite Fig. S5 in the Data Availability statement.	
Methods	
Please ensure that all information present in the Reporting Summary is also in the manuscript. This information is usually most appropriate in the Methods section.	OK
We allow unlimited space for Methods. The Methods must contain sufficient detail such that the work could be repeated. It is preferable that all key methods be included in the main manuscript, rather than in the Supplementary Information. Please avoid use of “as described previously” or similar in Line 427, 491, 495, 521, and instead detail the specific methods used with appropriate attribution.	We have revised it.
The Methods should include a separate section titled “Statistics and Reproducibility” with general information on how the statistical analyses of the data were conducted, and general information on the reproducibility of experiments (also those lacking statistical analysis), including the sample sizes and number of replicates and how replicates were defined. Please use the standard title and provide general info on the reproducibility.	We have revised it.
For studies using live vertebrates, a statement affirming that you have complied with all relevant ethical regulations for animal testing and research is necessary. A statement explicitly confirming if the study received ethical approval, including the name of the board and institution that approved the	The study was approved by the Ethics Committee of Beijing Shijitan Hospital Affiliated to Capital Medical University (2017-035).

study protocol is also required. The species, strain, sex and age of animals should be included.	6-week-old C57BL/6J male mice were purchased from Beijing Vital River Laboratory Animal Technology Co., Ltd. (Beijing, China).
Where human participants are involved, confirmation that all relevant ethical regulations were followed is needed, and that informed consent was obtained. This must be stated in the Methods section, including the name of the board and institution that approved the study protocol. Please confirm informed consent was obtained from patients.	The study was approved by the Ethics Committee of Beijing Shijitan Hospital Affiliated to Capital Medical University (2017-035) and registered on the Chinese Clinical Trial Registry (ChiCTR2100042049), as well as conducted under the guidelines of the Helsinki Declaration. We confirmed we obtained the informed consent from patients.
Data Policies	
The Data Availability statement must include:  ● Access details for deposited data, including repository name and unique data ID. ● How source data can be obtained. ● A statement that all other data are available from the corresponding author (or other sources, as applicable) on reasonable request. Note that ‘available upon request’ is only appropriate if immediate data access has not been mandated by our policies or by the editors. See here for more information about formatting your Data Availability Statement: http://www.springernature.com/gp/authors/research-data-	It is OK

policy/data-availability-statements/12330880	
Communications Biology has a strong preference for all data to be deposited in an approved repository. In some cases, data deposition may be required by the editor. We recommend the following data repositories:  ● GenBank (all DNA sequence data) ● NHGRI-EBI GWAS Catalog (GWAS summary statistics) ● PGS Catalog (polygenic risk scores) ● Gene Expression Omnibus (Microarray or RNA sequencing data) ● Sequence Read Archive (WGS or WES data) ● Protein Data Bank (protein structural data) ● OSF (neuroimaging raw data and EEG/EMG/MEG raw data) ● Neurovault (unthresholded statistical maps, parcellations, and atlases produced by MRI and PET studies) ● Image Data Resource (microscopy data) ● PRIDE (proteomics data) Data types without a specific repository can be deposited in a generalist repository, such as figshare or Dryad. For an up-to-date list of approved repositories, please visit https://www.springernature.com/gp/authors/research-data-policy/repositories/12327124.	We deposit the gut mycobiota analysis data on a public database (China National Microbiology Data Center (NMDC), under accession numbers NMDC10018246).
Data citation Please cite datasets stored in external repositories in the main reference list. For previously published datasets, we ask authors to cite both the related	There is no datasets stored in external repositories

research articles and the datasets themselves. For more information on how to cite datasets in submitted manuscripts, please see our data availability statements and data citations policy.	
End Notes	
Please check that your bibliography complies with the following:  ● Your bibliography should start with the heading “References”. The references must be numbered in the order of appearance in the text, then tables, then figures. ● Any in-text citations to references (e.g. "Gupta et al. show...") should be followed by their corresponding reference citation number from the reference list. ● Manuscript citations must include journal title, article title, volume number, page or article number or DOI, and year of publication. ● No publication can be present more than once in the reference list. ● No footnotes are permitted in the references or elsewhere. Text should be incorporated into the main text, the Methods section, or the Supplementary Information instead. ● Websites should only be listed in the references if they are in common use or curated. ● Where possible, preprints in the reference list should be updated with details of the published, peer-reviewed paper. ● Citations should be formatted in the text using superscript numbers. 	Yes
Please check that your 'Author Contributions' section individually lists the specific contribution of each author to the work. Each author must be referred to by name or initials. Where multiple authors possess identical initials, they must be clearly disambiguated from one another.	We have revised it

See our author contributions policy for further information: https://www.nature.com/nature-research/editorial-policies/authorship#author-contribution-statements	
No separate funding section is permitted. Please include your funding information in Acknowledgements instead.	We have revised it